# Examining Post-Training Quantization for Mixture-of-Experts: A Benchmark

## Abstract

Large Language Models (LLMs) have become foundational in the realm of natural language processing, demonstrating performance improvements as model sizes increase. The Mixture-of-Experts (MoE) approach offers a promising way to scale LLMs more efficiently by using fewer computational FLOPs through sparse activation. However, it suffers from significant memory overheads, necessitating model compression techniques. Post-training quantization, a popular method for model compression, proves less effective when directly applied to MoE models due to MoE's overlooked inherent sparsity. This paper explores several MoE structure-aware quantization heuristics, ranging from coarse to fine granularity, from MoE block to individual linear weight. Our investigations reveal critical principles: different MoE structures (*i.e.*, blocks, experts, linear layers) require varying numbers of weight bits for effective and efficient quantization. Conclusions are supported by extensive benchmarking across two representative MoE models and six tasks. We further introduce novel enhancements to more accurately identify the most critical weights in MoE quantization that necessitate higher bit allocations, including the linear weight outlier scorer and MoE block scorer. Additionally, subsequent experiments validate our findings in the context of both weight and activation quantization. Our code for reproducing all our experiments is provided as supplemental material.

## 1 Introduction

Large Language Models (LLMs) have achieved remarkable success in various natural language processing tasks, such as language understanding, reasoning, and generation, demonstrating superior performance and adaptability Brown et al. (2020); Jiang et al. (2023); Kaplan et al. (2020); OpenAI et al. (2024); Touvron et al. (2023). However, the rapid growth in model size, with state-of-the-art LLMs containing billions of parameters, poses significant challenges to computational resources and memory consumption Aminabadi et al. (2022); Lin et al. (2024); Shoeybi et al. (2020). The Mixture of Experts (MoE) Shazeer et al. (2017) architecture has emerged as a promising solution to address these challenges. MoE allows for the scaling up of LLMs while maintaining roughly constant FLOPs. By incorporating multiple expert networks and employing a sparse gating mechanism, MoE achieves efficient computation, enabling the development of larger models within the constraints of limited computational resources Dai et al. (2024); Fedus et al. (2022); Jiang et al. (2024).

Despite its advantages, MoE suffers from extensive memory costs, which hinder its practical deployment and widespread adoption. For example, the `Mixtral-8x7B` Jiang et al. (2024) MoE model takes around 180 GB memory while only 28 GB parameters are activated for each input token[1]. Model compression techniques tailored to MoE architectures are essential to address this issue. Existing MoE compression methods can be categorized into two main approaches: merging and pruning. Expert merging, such as MC-MoELi et al. (2024), aims to reduce the memory footprint by combining similar experts based on routing policy and compressing the resulting model using low-rank decomposition. On the other hand, expert pruning, such as task-specific pruning Chen et al. (2022), focuses on identifying and removing the least important experts or connections based on their contribution to a specific task. However, these approaches ① necessitate model retraining,

---

[1]This is evaluated in full precision (float32).

which is both extremely costly and time-consuming, particularly for state-of-the-art MoE LLMs of billion-size scale, and ② operate under task-specific settings, which limits their practicality for real-world applications.

Post-training quantization has emerged as a promising compression method widely applied to dense LLM models. Recent works, such as GPTQ Frantar et al. (2023a), which adapts quantization intervals based on the Hessian information, SmoothQuant Lin et al. (2024), which jointly quantizes the model weight and activation by offline migrating the activation outliers, have demonstrated the effectiveness of post-training quantization for LLMs toward 4 bits compression.

However, directly applying existing quantization methods to MoE models in a more extreme quantization setting, *e.g.* under 3 bits, leads to suboptimal results, potentially due to the overlooked sparsity nature of the MoE architecture. The sparse activation patterns and the dynamic routing mechanism in MoE pose unique challenges and opportunities for quantization, requiring novel approaches to utilize it effectively. The sparse expert activations in MoE models exhibit different statistical properties methodologies compared to dense activations, making conventional quantization methods difficult. Moreover, the dynamic routing mechanism, which selects a subset of experts for each input token, introduces additional complexity in terms of quantizing the routing weights and maintaining the sparsity pattern during inference. This yields the primary question to be explored:

*(Q) Can we leverage the sparsity nature of MoE architecture to establish more efficient and effective coarse-grained mixed-precision MoE quantization methods?*

To answer *(Q)*, we explore a wide range of MoE structure-aware quantization heuristics, ranging from coarse to fine granularity. We conduct a detailed comparative analysis of each of them, revealing critical principles: different MoE structures (*i.e.*, blocks, experts, linear layers) require varying numbers of weight bits for effective and efficient quantization. Extended from the gained insights, we propose methods to further improve the efficiency and effectiveness of mixed-precision quantization, including linear weight quantization scorer and MoE block quantization scorer.

In summary, our key contributions are listed below:

1. We establish the first benchmark for post-training quantization specifically designed for the Mixture-of-Experts architecture. This benchmark encompasses investigations into four critical MoE-related heuristics, evaluations across two MoE LLMs, six benchmark tasks, and a combination of both weight and activation quantization.

2. Our benchmark study uncovers a range of previously unexplored quantization principles and insights for MoE. These insights include empirical rules supporting optimal bit allocation strategies, highlighting the trade-offs such us those between attention and FFNN layers, and among different experts.

3. Leveraging the insights from our benchmark study, we introduce novel enhancements to improve existing heuristics. These include the development of linear-weight and MoE block scorers to identify the most critical components of the MoE model, thereby guiding more effective quantization bit assignments.

## 2 RELATED WORKS

**Mixture-of-Experts.** The Mixture-of-Experts (MoE) approach Shazeer et al. (2017) enhances neural network scalability by using router networks to activate model segments according to input tokens selectively. As the dominant architecture in NLP, numerous efforts have adapted feed-forward neural networks (FFNNs) within Transformers to incorporate MoE layers, constructing MoE language modelsDai et al. (2024); Fedus et al. (2022); Jiang et al. (2024). Additionally, several variants of the standard MoE architecture exist. For example, DeepSeek-MoE Dai et al. (2024) employs numerous finely segmented experts and designates a select few as shared experts to capture common knowledge. MoE's application in LLMs is widely acknowledged for its superior generative abilities and remarkable computing efficiency Artetxe et al. (2022); Dai et al. (2024); Fedus et al. (2022); Jiang et al. (2024); Krajewski et al. (2024); Rajbhandari et al. (2022). The recent work Mixtral Jiang et al. (2024) illustrates that MoE can match the performance of equivalent full-parameter LLMs while utilizing far fewer active parameters. However, MoE suffers from significant memory overhead issues, posing challenges to its efficient deployment Li et al. (2024).

**MoE Compression.** MoE models benefit from reduced FLOPs but are constrained by their significant memory overhead. Current works to reduce the memory overhead of MoE models mainly focus on reducing the number of experts. An earlier approach Chen et al. (2022) involves pruning non-essential experts for a specific downstream task during fine-tuning, utilizing statistics based on cumulative usage frequency. Another method, MC-SMoE Li et al. (2024), introduces a pipeline that identifies and groups similar experts, subsequently merging them and further decomposing the merged expert into low-rank components within each group. However, these approaches are developed under task-specific fine-tuning settings and do not explore the development of the MoE compression towards a general post-training model.

**Post-Training Quantization.** Post-training quantization reduces computational and storage demands by converting pre-trained models from high-precision to lower-precision formats without extensive retraining Frantar et al. (2023b;a). It has been widely applied to LLMs, optimizing them for deployment on resource-constrained devices. Techniques like layer-wise quantization and mixed-precision schemes are designed for minimal performance degradation while reducing model size and computational requirements efficiently Liu et al. (2023); Pan et al. (2023); Sharify et al. (2024). Recent methods such as SmoothQuant Xiao et al. (2024), GPTQ Frantar et al. (2023a), AWQ Lin et al. (2024), and address specific challenges for LLMs. SmoothQuant Xiao et al. (2024) ensures smooth precision transitions across layers, reducing quantization errors and maintaining performance. GPTQ Frantar et al. (2023a) employs layer-wise and mixed-precision quantization to balance efficiency and accuracy. AWQ Lin et al. (2024) adapts to weight sensitivity, preserving critical weights' precision while aggressively quantizing less sensitive ones. These advancements in PTQ enable significant reductions in computational and storage requirements while preserving LLM performance.

## 3 Reviewing Quantization and MoE

### 3.1 Quantization Method

The primary objective of this work is to benchmark several MoE-related heuristics combined with established LLM quantization techniques. Given that the substantial memory overhead of MoE models predominantly originates from their weights, we adopt GPTQ Frantar et al. (2023a), a popular weight quantization method. GPTQ executes layer-by-layer weight quantization by addressing a specific reconstruction problem for each layer. Specifically, let $\mathbf{W}$ represent the weights of a linear layer and $\mathbf{X}$ denote the input to that layer derived from a small subset of calibration data, the reconstruction problem is defined as follows:

$$\text{argmin}_{\widehat{\mathbf{W}}}, ||\mathbf{W}\mathbf{X} - \widehat{\mathbf{W}}\mathbf{X}||_2^2. \tag{1}$$

This objective, being the sum of squared errors, forms a quadratic equation, allowing the greedy-optimal update of weights to be calculated element-by-element using the Hessian information, $\mathbf{H} = 2\mathbf{X}\mathbf{X}^\top$. GPTQ further enhances this process by incorporating a lazy-batch update and a Cholesky reformulation, to improve scalability and numerical stability for LLM quantization.

### 3.2 Mixture-of-Experts

There are several variants of MoE in the context of LLMs, such as attention MoE and FFNN MoE. In this work, we explore the quantization of MoE models that utilize router networks to selectively activate FFNNs for different input tokens. Specifically, for the $i$-th expert's feed-forward function at the $l$-th transformer layer, denoted as $\text{FFNN}i^l(\cdot)$, the output of the MoE layer for the input hidden states $\mathbf{X}$ is given by:

$$\text{FFNN}_{\text{MoE}}^l(\mathbf{X}) = \sum_{i=1}^{l} \mathcal{G}(\mathbf{W}_l\mathbf{X}) \cdot \text{FFNN}_i^l(\mathbf{X}), \tag{2}$$

where $\mathbf{W}_l$ represents a linear routing matrix and $\mathcal{G}(\cdot)$ is a routing function that typically employs a top-$k$ selection mechanism, resulting in a sparse output. Due to the duplication of FFNN layers, the principal memory overhead in the MoE model is attributed to the FFNN component.

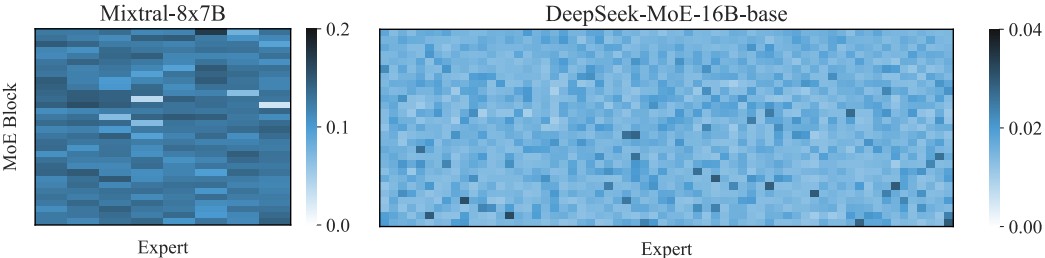

Figure 1: Visualization of expert usage of the two MoE models used in this work. It is evaluated on the quantization calibration data, *i.e.*, 512 random 4096 token sequences from the WikiText dataset Merity et al. (2016).

### 3.3 EXPERT USAGE AS A HEURISTIC

As the routing of experts in MoE models is not ideally balanced, expert usage frequency and its variants have emerged as prevalent heuristics for measuring the importance of different experts within an MoE block Chen et al. (2022); Li et al. (2024). For instance, task-specific expert pruning proposed by Chen et al. (2022) uses a criterion based on cumulatively calculated expert routing probabilities for pruning during fine-tuning on a specific task. In this paper, focusing on post-training quantization, we utilize the routing distribution from the calibration data as the heuristic for expert usage. Specifically, for the $l$-th MoE block, equipped with a routing matrix $\mathbf{W}l \in \mathbb{R}^{e \times d}$ and input hidden states $\mathbf{X} \in \mathbb{R}^{b \times d}$ from the calibration data, the expert usage heuristic is calculated as follows:

$$\texttt{usage} = \texttt{normalize}\left(\sum_i \mathcal{G}(\mathbf{W}_l \mathbf{X}_i)\right), \tag{3}$$

where $\mathcal{G}(\cdot)$ is the routing function employing a top-$k$ selection mechanism that yields a sparse binary output. We visualize the calculated expert usage of `Mixtral-8x7B` and `DeepSeek-MoE-16B-base` MoE models on the quantization calibration data, as shown in Figure 1. Note that `Mixtral-8x7B` demonstrates a more balanced routing distribution than `DeepSeek-MoE-16B-base`.

## 4 BENCHMARK POST-QUANTIZATION METHODS FOR MOE

In this section, we present several heuristics for MoE quantization and the empirical performance of them. Our benchmarking covers two MoE models and six popular tasks.

### 4.1 BENCHMARK SETUPS

**MoE Models.** We select two representative MoE models for our benchmark evaluation, *i.e.*, `Mixtral-8x7B` Jiang et al. (2024) and `DeepSeek-MoE-16B-base` Dai et al. (2024). `Mixtral-8x7B` substitutes every FFNN with a MoE block and has 8 experts per MoE block with top-2 routing, while `DeepSeed-MoE-16B-base` uses a fine-grained MoE architecture by including 64 experts with top-6 routing and 2 shared experts per MoE block. Notably, the `DeepSeek-MoE-16B-base` model incorporates a dense architecture in its first transformer block while employing an MoE architecture in subsequent blocks for better training stability.

**Quantization.** We mainly focus on *weight-only grouped mixed-precision* quantization, though we also extend our experiments and conclusions to its combination with activation quantization in Section 5. The weight-only experiments utilize GPTQ Frantar et al. (2023a), while those that combine weight and activation quantization utilize SmoothQuant Xiao et al. (2024), without loss of generality. Throughout this work, we use a group size of 128. Our experiments emphasize an extreme quantization scenario, where most weights are quantized to either 2 or 4 bits.

**Calibration and Evaluation Details.** We use the calibration data consisting of 512 random 4096 token sequences from the WikiText dataset Merity et al. (2016), following GPTQ Frantar et al. (2023a). Unlike previous literature that focuses on language modeling benchmarks Xiao et al. (2024); Lin et al. (2024); Frantar et al. (2023a), we evaluate all the methods on six popular LLM tasks for a practical benchmarking: WinoGrande ai2 (2019), COPA Gordon et al. (2012), OpenBookQA (OBQA) Mihaylov et al. (2018), HellaSwag Zellers et al. (2019), and MMLU Hendrycks et al. (2021). We report the performance on MMLU with 5-shot and all others with zero-shot. All ex-

periments are conducted with PyTorch on 3 NVIDIA H100, and we utilize *lm-evaluation-harness* [2] for the evaluation of all tasks.

## 4.2 BENCHMARK RESULTS

We first evaluate several MoE heuristics quantization methods based on GPTQ on `Mixtral-8x7B` and `DeepSeek-MoE-16B`. We present our benchmark conclusions by answering the following research questions.

**Q1: Is expert usage frequency a good quantization heuristic? A: Fairly good.** Expert usage frequency is a popular heuristic in the compression of MoE models, predicated on the insight that less frequently used experts are likely less crucial. Our experiments, detailed in Table 1, corroborate its effectiveness as a quantization heuristic for MoE models. In particular, for the `DeepSeek-MoE-16B-base` model, this heuristic markedly outperforms the strategy of randomly allocating more bits to experts, likely due to the model's unbalanced routing distribution. However, with the `Mixtral-8x7B` model, where the routing distribution is more balanced, the advantage of using expert usage frequency over random allocation is less significant.

Table 1: Comparison of the expert usage frequency heuristic *v.s.* random allocation. For the `Mixtral-8x7B` model, we compare the allocation of 4 bits to the top-{2, 4} most frequently used experts per MoE block against randomly selecting {2, 4} experts for the same bit allocation. For the `DeepSeek-MoE-16B-base` model, we keep shared expert {8} bits and compare between top-{10, 15, 20, 25} most frequently used experts against randomly selecting {10, 15, 20, 25} experts per MoE block. The remaining experts are quantized to 2 bits, while all attention layers are uniformly quantized to 4 bits. All random experimental results in the format of $a \pm b$ provide the mean value $a$ and its standard deviation $b$ over 3 independent trials.

| Methodology | Bits | WinoGrande (%) | COPA (%) | OBQA (%) | HellaSwag (%) | PIQA (%) | MMLU (%) | Average (%) |
|---|---|---|---|---|---|---|---|---|
| | | | | Mixtral-8x7B | | | | |
| Random 2 | 2.54 | $\mathbf{58.59 \pm 2.57}$ | $68.00 \pm 11.27$ | $\mathbf{33.00 \pm 1.78}$ | $46.60 \pm 18.21$ | $60.14 \pm 9.32$ | $28.26 \pm 4.64$ | $49.10 \pm 7.73$ |
| Frequent 2 | 2.54 | 58.33 | **76.00** | 32.00 | **56.62** | **66.21** | **36.01** | **54.20** |
| Random 4 | 3.03 | $67.77 \pm 0.36$ | $\mathbf{86.33 \pm 3.51}$ | $38.47 \pm 0.31$ | $67.48 \pm 0.52$ | $\mathbf{73.99 \pm 0.52}$ | $48.13 \pm 2.57$ | $63.70 \pm 0.49$ |
| Frequent 4 | 3.03 | **68.82** | 86.00 | **38.80** | **67.68** | 72.20 | **49.42** | **63.82** |
| | | | | DeepSeek-MoE-16B-base | | | | |
| Random 10 | 2.53 | $\mathbf{67.28 \pm 0.04}$ | $88.50 \pm 1.50$ | $38.40 \pm 0.80$ | $\mathbf{70.99 \pm 0.50}$ | $\mathbf{76.74 \pm 0.84}$ | $35.23 \pm 0.09$ | $62.86 \pm 0.60$ |
| Frequent 10 | 2.53 | 66.46 | 87.00 | **39.60** | 70.31 | 76.71 | **37.84** | **62.99** |
| Random 15 | 2.68 | $\mathbf{67.25 \pm 0.47}$ | $84.50 \pm 2.50$ | $\mathbf{40.00 \pm 0.60}$ | $\mathbf{71.79 \pm 0.43}$ | $76.85 \pm 0.08$ | $35.71 \pm 0.82$ | $62.68 \pm 0.71$ |
| Frequent 15 | 2.68 | 67.17 | **88.00** | 39.00 | 71.09 | **76.93** | **40.59** | **63.80** |
| Random 20 | 2.83 | $67.25 \pm 0.47$ | $84.50 \pm 2.50$ | $40.00 \pm 0.60$ | $71.79 \pm 0.43$ | $76.85 \pm 0.08$ | $35.71 \pm 0.82$ | $62.68 \pm 0.71$ |
| Frequent 20 | 2.83 | **67.25** | **86.00** | **40.40** | **72.06** | **77.58** | **40.78** | **64.01** |
| Random 25 | 2.97 | $67.72 \pm 0.24$ | $89.00 \pm 1.00$ | $\mathbf{40.70 \pm 0.10}$ | $71.98 \pm 0.19$ | $77.04 \pm 0.05$ | $36.54 \pm 1.55$ | $63.83 \pm 0.04$ |
| Frequent 25 | 2.97 | **67.72** | **90.00** | 39.20 | **72.83** | **77.15** | **41.06** | **64.66** |

**Q2: Attention *vs.* FFNN: Which Deserves More Bits in MoE? A: Attention layers are more bit-efficient.** Because of the unique characteristics of the feedforward neural network (FFNN) within the mixture of experts (MoE) framework. we explore the attention layer and the feedforward neural network layer, which deserves more bits. We compare the performance evaluated by quantizing the attention layers with more bits *v.s.* randomly selecting experts in the FFNN layers with more bits, maintaining the same average bits of the entire MoE model for a fair comparison. Specifically, we quantize the attention weight or randomly selected FFNN weight to {2, 4, 8} bits, while All other weights are quantized to 2 bits by default. As illustrated in Figure 2, quantizing attention weights to higher bit levels (*i.e.*, 4 or 8 bits) consistently results in significant performance gains (over 5%) under each average bit allocation for the MoE model. This greater efficiency likely stems from the fact that attention weights are activated for every token,

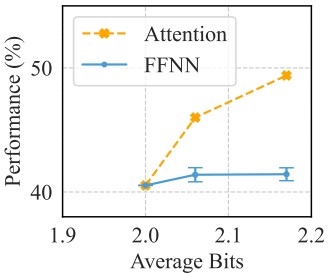

Figure 2: Comparison of quantizing more bits for attention *vs.* FFNN. It is evaluated on the `Mixtral-8x7B` model. FFNN results show the mean and standard deviation (error bars) from 3 independent trials.

[2]https://github.com/EleutherAI/lm-evaluation-harness

while FFNN weights only engage with a subset of the input tokens. Consequently, increasing the quantization bits for FFNN weights does not benefit all inputs. Based on these findings, attention weights are quantized to $4$ bits by default in all following experiments.

Table 2: Comparison between quantizing first $k$ *v.s.* last $k$ MoE blocks with higher (*i.e.* $4$) bits. All weights in attention layers are quantized to $4$ bits, and the other weights are quantized to $2$ bits. In `DeepSeek-MoE-16B-base` model, we keep the first block that is dense block as $4$ bits by default. We evaluate $k$ of $4$ and $8$. The higher performance of each comparison pair is marked as **bold**.

| Methodology | Bits | WinoGrande (%) | COPA (%) | OBQA (%) | HellaSwag (%) | PIQA (%) | MMLU (%) | Average (%) |
|---|---|---|---|---|---|---|---|---|
| | | | | `Mixtral-8x7B` | | | | |
| First 4 | 2.30 | **57.85** | **72.00** | **32.80** | **52.80** | **61.59** | **29.65** | **51.12** |
| Last 4 | 2.30 | 53.75 | 60.00 | 27.80 | 46.25 | 58.87 | 26.56 | 45.54 |
| First 8 | 2.54 | **62.11** | **85.00** | **35.80** | **62.72** | **67.74** | **35.61** | **58.16** |
| Last 8 | 2.54 | 52.09 | 69.00 | 29.60 | 47.87 | 59.58 | 26.03 | 47.36 |
| | | | | `DeepSeek-MoE-16B-base` | | | | |
| First 4 | 2.29 | **65.27** | **85.00** | **38.40** | **64.42** | 72.74 | **28.88** | **59.12** |
| Last 4 | 2.29 | 62.90 | 83.00 | 36.00 | 64.41 | **74.65** | 27.38 | 58.06 |
| First 8 | 2.63 | **64.09** | **86.00** | **38.75** | **67.84** | 75.35 | 30.12 | **60.36** |
| Last 8 | 2.63 | 62.83 | 83.00 | 37.80 | 65.94 | **75.73** | **31.00** | 59.38 |

**Q3: Do the model's first or last MoE blocks deserve more bits in quantization? A: The first MoE blocks.** As more and more Mixture-of-Experts (MoE) architectures emerge, we investigate which layer of the MoE block is more critical and thus deserves more bits during the quantization process. As shown in Table 2, we evaluate the performance of allocating more bits to the first $k$ blocks versus the last $k$ blocks in quantization. The results consistently indicate that higher bit quantization of the first few blocks yields better performance, suggesting that we can allocate more bits to the quantization of the first blocks of the model. This observation aligns with prior studies that have empirically confirmed the greater importance of the first few Transformer blocks Dai et al. (2024); Ma et al. (2023).

**Q4: Does the shared expert always deserve more bits? A: Yes.** The `DeepSeek-MoE-16B-base` model includes two shared experts within each MoE block to obtain common knowledge across varying domains and alleviate the parameter redundancy. To evaluate their role in quantization, we compare quantizing these two shared experts with more bits *v.s.* randomly selecting two non-shared experts for more bit allocation, maintaining the same average bits for a fair comparison. The shared or random non-shared experts are quantized to 2, 4, 8 bits, while attention weights are set to $4$ bits and all other weights to $2$ bits. As depicted in Figure 3, allocating higher bit levels (*i.e.*, $4$ or $8$ bits) to shared experts consistently yields superior performance. This enhanced efficiency and effectiveness are attributed to the shared experts being activated for every input token, unlike non-shared experts, which only engage with specific subsets of the tokens. Allocating more quantization bits to shared experts thus proves to be both more efficient and effective.

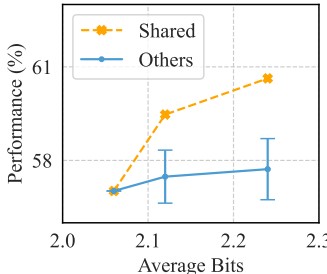

Figure 3: Comparison of quantizing more bits for shared experts *vs.* others experts. "Others" results show the mean and standard deviation from 3 independent trials of random selecting 2 experts from the non-shared experts.

## 5 EXTENDED STUDY TO IMPROVE MoE QUANTIZATION

In this section, we expand our benchmark results from weight quantization to include activation quantization. Additionally, we introduce two novel algorithmic advancements aimed at enhancing the effectiveness of identifying crucial components within MoE models for improved quantization performance.

## 5.1 QUANTIZING BOTH WEIGHT AND ACTIVATION

We further expand our study by simultaneously including weight and activation quantization to validate our conclusions. Specifically, we employ SmoothQuant Xiao et al. (2024) combined with our expert-usage-frequency heuristic. It selects the top-2 experts' weights per MoE block in the `Mixtral-8x7B` model and the top-16 experts' weights per MoE block in the `DeepSeek-MoE-16B-base` for quantization to 4 bits, while quantizing all other weights to 2 bits. The evaluation results, presented in Table 3, reveal the marginal performance gap across different activation quantization bits. This demonstrates that our conclusions regarding weight quantization are robust and can be reliably extended to various activation quantization scenarios as well.

Table 3: Combination of activation quantization with the expert-usage-based heuristic. We evaluate it on the top-2 most frequently used experts per MoE block in `Mixtral-8x7B` and the top-16 frequent experts per MoE block in `DeepSeek-MoE-16B-base`, quantizing these experts to 4 bits. All attention weights are also quantized to 4 bits, while all other weights are quantized to 2 bits. The higher performance of each comparison pair is marked as **bold**.

| Weight Bits | Activation Bits | WinoGrande (%) | COPA (%) | OBQA (%) | HellaSwag (%) | PIQA (%) | MMLU (%) | Average (%) |
|---|---|---|---|---|---|---|---|---|
| | | | | Mixtral-8x7B | | | | |
| | 4 | **50.28** | 51.00 | **26.80** | 25.99 | **51.90** | 23.85 | 38.30 |
| 2.54 | 8 | 50.04 | **60.00** | **26.80** | **26.55** | 51.58 | 23.77 | **39.79** |
| | 16 | 49.41 | **60.00** | 26.60 | 26.53 | 51.85 | **23.86** | 39.71 |
| | | | | DeepSeek-MoE-16B-base | | | | |
| | 4 | 48.22 | **53.00** | 27.20 | 26.12 | 50.65 | **26.86** | 38.67 |
| 2.71 | 8 | 49.96 | 51.00 | **27.60** | **26.58** | **53.86** | 25.91 | **39.15** |
| | 16 | **50.19** | 51.00 | **27.60** | 26.43 | 53.70 | 25.16 | 39.01 |

## 5.2 CONCENTRATING LINEAR LAYERS WITH LARGER WEIGHT OUTLIERS

**Insight.** From the quantization perspective, the larger the range of a weight magnitude group, the more difficult it will be for quantization. We found that, in MoE, each FFNN linear weight matrix consists predominantly of values within a narrow range, interspersed with a few significant outliers. Consequently, we propose a weight-magnitude-based metric to identify those linear layers that are challenging to quantize effectively, thereby necessitating a higher allocation of quantization bits.

**Methodology.** We define the metrics to estimate the outliers of weights by the maximum ratio of the largest to the average absolute magnitude within each column. Specifically, for a weight matrix $\mathbf{W} \in \mathbb{R}^{m \times n}$, we compute the metric `outlier-score`$(\mathbf{W})$ as follows:

$$\texttt{outlier-score}(\mathbf{W}) = \max_j \left( \frac{\max(|\mathbf{W}:, j|)}{\text{mean}(|\mathbf{W}:, j|)} \right), \tag{4}$$

where $|\mathbf{W}:, j|$ is the absolute value of $\mathbf{W}$'s $j$-th column. With this metric, we can identify those linear layers that require more quantization bits and allocate more to them, providing an effective trade-off between performance and efficiency. The overall procedure is detailed in Algorithm 1.

---

**Algorithm 1** The Procedure of MoE Mixed-Precision Quantization with `outlier-score`.

---

1: **Initialize:** A MoE model with $l$ linear layers across all the FFNN experts, the number of linear layers for 4 bit quantization $k$.
2: Let $\mathcal{M}$ and $\mathcal{S}$ represent the set of each linear layer matrix in FFNN and its score, respectively.
3: **for** linear layer $i = 1, \ldots, l$ **do**
4: $\quad \mathbf{W} \leftarrow \mathcal{M}[i]$
5: $\quad \mathcal{S}[i] \leftarrow \max_j \left( \frac{\max(|\mathbf{W}:, j|)}{\text{mean}(|\mathbf{W}:, j|)} \right)$
6: **end for**
7: $\alpha \leftarrow \texttt{sorted}(\mathcal{S})[k]$
8: $\texttt{4bits-quantize}(\{\mathcal{M}[i] \mid \mathcal{S}[i] >= \alpha\})$
9: $\texttt{2bits-quantize}(\{\mathcal{M}[i] \mid \mathcal{S}[i] < \alpha\})$
10: **Return:** A quantized mixed-precision MoE model.

---

**Experiments.** We evaluate this metric by comparing its application for the top-$p\%$ of linear layers against randomly selecting linear layers, using percentages of $25\%$ and $50\%$. In `DeepSeek-MoE-16B-base` model, we also involve shared experts using this metric. As illustrated in Table 4, our proposed scorer consistently outperforms the random baseline on both models and almost all tasks (except HellaSwag and MMLU). This is particularly evident in the `DeepSeek-MoE-16B-base` model, where it achieves an average performance improvement of about $3\%$, aligning with our expectations.

Table 4: Comparison between using our linear weight scorer *vs.* random selection of linear layers for bit allocation in quantization. We evaluate by quantizing $25\%$ of the linear layers across all MoE blocks (*i.e.*, FFNN) to $4$ bits. All attention weights are quantized to $4$ bits, and all other weights are quantized to $2$ bits. In each comparison pair, the higher performance is highlighted in **bold**. All random experimental results in the format of $a \pm b$ provide the mean value $a$ and its standard deviation $b$ over 3 independent trials.

| Methodology | Bits | WinoGrande (%) | COPA (%) | OBQA (%) | HellaSwag (%) | PIQA (%) | MMLU (%) | Average (%) |
|---|---|---|---|---|---|---|---|---|
| | | | | `Mixtral-8x7B` | | | | |
| Random 25% | 2.54 | $60.74 \pm 0.63$ | $78.67 \pm 4.62$ | $34.07 \pm 1.63$ | $\mathbf{57.36 \pm 0.53}$ | $68.19 \pm 0.74$ | $\mathbf{32.49 \pm 1.60}$ | $55.25 \pm 0.95$ |
| Ours top-25% | 2.54 | **62.19** | **83.00** | **35.80** | 57.04 | **68.23** | 30.95 | **56.20** |
| | | | | `DeepSeek-MoE-16B-base` | | | | |
| Random 25% | 2.54 | $64.04 \pm 0.78$ | $84.67 \pm 4.73$ | $37.53 \pm 0.46$ | $67.39 \pm 0.71$ | $74.61 \pm 0.60$ | $29.43 \pm 1.31$ | $59.61 \pm 0.76$ |
| Ours top-25% | 2.54 | **66.14** | **85.00** | **38.80** | **71.65** | **76.82** | **36.19** | **62.43** |

**Visualization.** As shown in Figure 4, we visualize the proposed `outlier-score` for each FFNN linear weight within the `Mixtral-8x7B` model. Given that each FFNN expert includes three linear layers, namely the *gate projection*, *up projection*, and *down projection*, we visualize these components separately to ensure clarity. Notably, many of the *down projection* linear layers, particularly those positioned later in the MoE model, exhibit significantly higher `outlier-scores` compared to others.

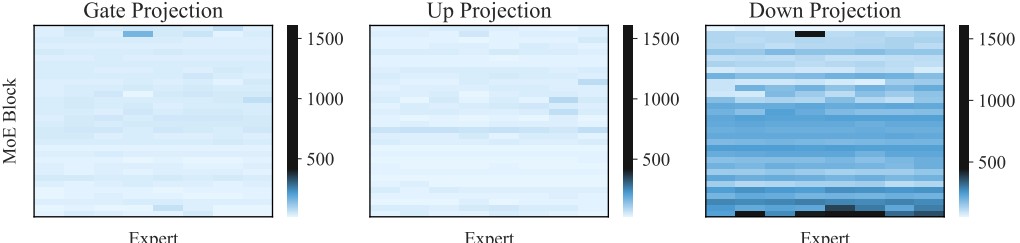

Figure 4: Visualization of the `outlier-score` metric applied to each FFNN linear weight matrix within the `Mixtral-8x7B` model. For clearer visualization, we present separate components, including the *gate projection* (left), *up projection* (middle), and *down projection* (right) in FFNN experts.

### 5.3 TRAINING BLOCK QUANTIZATION IMPORTANCE SCORE PREDICTOR.

Inspired by Q3 in Section 4.2, which demonstrates that allocating more bits to different MoE blocks yields variable performance improvements, we propose a novel method to identify and quantize those critical blocks with additional bits. Specifically, this section outlines our approach to calculating importance scores for bit allocation using a data-driven method with a lightweight predictor.

**Insight.** We find an increasing cosine similarity between the tensors generated before and after the FFN blocks for some of the MoE blocks, indicating less important computation results produced by these blocks. This observation also aligns with observations on dense models in previous literature Jaiswal et al. (2024). Therefore, the basic idea is that less accurate output of these blocks producing tokens with high cosine similarity will not affect the overall model performance much, thus lower weight bits might not hurt performance much.

**Methodology.** To capture the generalized hidden states' dynamic information of each MoE block, we train a small two-layer FFNN with a tangent activation function. This network predicts the cosine similarity between the input and output hidden states. We utilize a dataset of $400$ random sequences, each containing $1024$ tokens from the WikiText dataset Merity et al. (2016), for training. The detailed training procedure is in Algorithm 2. During quantization, we employ this predictor

to run inference on the calibration data, computing the average predicted score for each MoE block across all tokens. A higher predicted score indicates less important and fewer bits for quantization.

---

**Algorithm 2** The Training Procedure of Block Score Predictor.

---

1: **Initialize:** A MoE block $M$, token input and output embedding set at block $M$ $\{(\mathbf{x}_i, \mathbf{y}_i)\}_{i \in [N]}$.
2: Let $\mathcal{BSP}$ denotes the block score predictor.
3: $\mathcal{X} \leftarrow \{\mathbf{x}_i \mid i \in [N]\}$
4: $\mathcal{S} \leftarrow \{\texttt{cosine}(\mathbf{x}_i, \mathbf{y}_i) \mid i \in [N]\}$
5: $\mathcal{BSP} \leftarrow \texttt{train}(\mathcal{X}, \mathcal{S})$
6: **Return:** The importance score predictor $\mathcal{BSP}$ for MoE Block $M$.

---

**Experiments.** In Table 5, we compare the performance of using our block importance predictor to select $k$ MoE blocks for 4 bits and others for 2 bits quantization with two other baselines: ① random selecting $k$ MoE blocks, and ② first $k$ MoE blocks (as it is the best in Q3 in Section 4.2). Evaluation results on the `DeepSeek-MoE-16B-base` model are presented in Table 5, showing the superiority of our method against the other two baselines.

Table 5: Comparison between using our MoE block importance predictor *v.s.* two baselines: ①random selecting and ②first $k$ MoE blocks. The predicted or selected MoE blocks are quantized to 4 bits, all attention weights are quantized to 4 bits, and all other weights are quantized to 2 bits. In each comparison, the highest performance is highlighted in **bold**. All random experimental results in the format of $a \pm b$ provide the mean value $a$ and its standard deviation $b$ over 3 independent trials.

| Methodology | Bits | WinoGrande (%) | COPA (%) | OBQA (%) | HellaSwag (%) | PIQA (%) | MMLU (%) | Average (%) |
|---|---|---|---|---|---|---|---|---|
| | | | | DeepSeek-MoE-16B-base | | | | |
| Random 4 | 2.29 | $61.09 \pm 0.78$ | $83.00 \pm 0.00$ | $37.20 \pm 0.85$ | $64.88 \pm 0.30$ | $74.21 \pm 0.08$ | $27.82 \pm 0.46$ | $58.03 \pm 0.13$ |
| First 4 | 2.29 | 65.27 | **85.00** | **38.40** | 64.42 | 72.74 | 28.88 | 59.12 |
| Predicted 4 | 2.29 | **65.27** | 83.00 | 36.60 | **64.88** | **74.54** | **37.75** | **60.34** |
| Random 8 | 2.63 | $64.48 \pm 0.83$ | $85.33 \pm 3.21$ | $38.73 \pm 0.95$ | $67.57 \pm 0.40$ | **$75.43 \pm 0.14$** | **$31.41 \pm 2.17$** | $60.49 \pm 0.56$ |
| First 8 | 2.63 | 64.09 | **86.00** | **38.75** | 67.84 | 75.35 | 30.12 | 60.36 |
| Predicted 8 | 2.63 | **65.35** | **86.00** | 38.00 | **68.77** | 75.35 | 30.01 | **60.58** |
| Random 12 | 2.92 | $64.64 \pm 0.89$ | $83.50 \pm 0.71$ | **$39.60 \pm 2.83$** | $69.51 \pm 0.56$ | $75.98 \pm 0.42$ | $32.57 \pm 0.30$ | $60.97 \pm 0.62$ |
| First 12 | 2.92 | 67.48 | **88.00** | 38.60 | 70.59 | 75.95 | **39.25** | 63.31 |
| Predicted 12 | 2.92 | **68.11** | **88.00** | 39.20 | **71.82** | **76.66** | 38.45 | **63.71** |

**Visualization.** We visualize the predicted scores of each MoE block using our trained predictors in the `DeepSeek-MoE-16B-base` model, as shown in Figure 5. Notably, MoE blocks situated in the middle of the model, which exhibit higher scores, are regarded as less critical. Consequently, these blocks will be quantized with fewer bits (specifically, 2 bits), reflecting their lower importance. Besides, Figure 5 also demonstrates that the first few MoE blocks are more important aligned with Q3. Interestingly, the last two blocks of the `DeepSeek-MoE-16B-base` model are also crucial, thereby allocating more bits and yielding better performance.

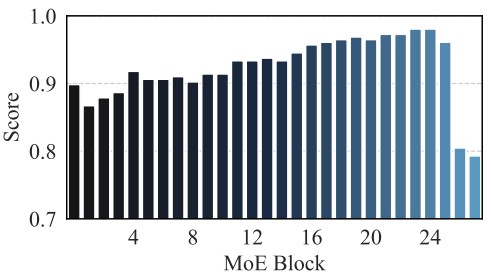

Figure 5: Visualization of the predicted MoE block importance score using our trained predictors.

## 6 CONCLUSION

This work investigates various heuristic-based MoE quantization methods in the post-training setting. While vanilla quantization techniques (*e.g.*, GPTQ) prove less effective and efficient when applied directly to MoE models, determining which MoE model components should be allocated more quantization bits remains an open question. We present the first benchmark study on MoE quantization, revealing critical heuristic-based principles, such as the importance disparities among different MoE blocks. Drawing on these insights, we introduce innovative techniques, including a block importance predictor and a linear layer outlier range scorer, to more precisely identify components that benefit from increased bit quantization. These methods substantially improve the quantization process's effectiveness and efficiency for MoE models.

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

# A APPENDIX

## A.1 EVALUATION DATASETS

In this section, we introduce details of the datasets in our evaluation. For a more comprehensive study, we have selected six popular benchmark tasks: WinoGrande, COPA, OpenBookQA (OBQA), HellaSwag, and MMLU.

**WinoGrande** ai2 (2019) is a large-scale dataset designed for commonsense reasoning, consisting of pronoun resolution problems. Each instance in the dataset presents a sentence with an ambiguous pronoun that needs to be resolved based on context. This task tests the model's ability to understand and reason about everyday situations.

**The Choice of Plausible Alternatives (COPA)** dataset Gordon et al. (2012) focuses on causal reasoning. Each question in COPA consists of a premise and two choices, where the model must select the more plausible alternative. This task evaluates the model's understanding of cause-and-effect relationships in natural language.

**OpenBookQA** Mihaylov et al. (2018) is a multiple-choice question-answering dataset that requires the model to use both scientific facts and commonsense knowledge. The dataset challenges the model's ability to combine factual knowledge with reasoning to answer questions correctly.

**HellaSwag** Zellers et al. (2019) is a benchmark for commonsense NLI (Natural Language Inference) that tests the model's ability to predict the most plausible continuation of a given sentence. The dataset contains scenarios from various domains, such as cooking and sports, requiring the model to understand context and plausibility.

**The Massive Multitask Language Understanding (MMLU)** benchmark Hendrycks et al. (2021) evaluates models across a wide range of subjects, from elementary mathematics to law. For this study, we report performance on MMLU with a 5-shot setting, where the model is given five examples per task before evaluation, allowing us to gauge the model's few-shot learning capabilities.

We perform a zero-shot evaluation on WinoGrande, COPA, OpenBookQA, and HellaSwag, where the model is not provided with any task-specific training examples. For MMLU, a 5-shot evaluation protocol is adopted, providing five examples per task. This setup helps us assess the generalization ability of the models across different types of reasoning and knowledge-based tasks.

## A.2 RANDOM SEED

For all the random selection experiments, we use random seeds $\{42, 43, 44\}$ to conduct three independent trials and then report the standard deviation and mean.

## A.3 Further Discussion

In this section, we present further discussion of the `DeepSeek-MoE-16B-base` performance across different bits.

**Expert usage frequency.** As shown by Q1 in Section 4.2, expert usage frequency is a critical metric in the compression of MoE models, predicated on the insight that less frequently used experts are likely less crucial. We present further discussion of ablation on the bits allocation in the expert-frequency-based methods.

Table 6: Ablation on the allocated bits for the selected top-$k$ experts based on frequency. We compare the allocation of $\{4, 8\}$ bits of the top-$k$ experts based on frequency, and all other experts are quantized to 2 bits.

| Top | Top-$k$ bits | Bits | WinoGrande (%) | COPA (%) | OBQA (%) | HellaSwag (%) | PIQA (%) | MMLU (%) | Average (%) |
|---|---|---|---|---|---|---|---|---|---|
| 1 | 4 | 2.29 | 66.30 | 83.00 | 39.00 | 69.28 | 75.03 | 35.02 | 61.27 |
| | 8 | 2.35 | 66.14 | 87.00 | 39.80 | 69.44 | 75.30 | 34.04 | 61.95 |
| 2 | 4 | 2.32 | 66.38 | 88.00 | 38.60 | 69.44 | 76.06 | 36.49 | 62.49 |
| | 8 | 2.44 | 65.98 | 90.00 | 38.60 | 69.77 | 76.33 | 35.82 | 62.75 |
| 5 | 4 | 2.41 | 66.54 | 87.00 | 38.40 | 70.13 | 76.12 | 38.02 | 62.70 |
| | 8 | 2.70 | 64.96 | 89.00 | 39.40 | 70.56 | 75.90 | 38.56 | 63.06 |
| 10 | 4 | 2.55 | 67.17 | 86.00 | 39.20 | 70.55 | 76.55 | 39.11 | 63.10 |
| | 8 | 3.14 | 66.06 | 88.00 | 39.00 | 70.81 | 76.71 | 39.30 | 63.31 |
| 15 | 4 | 2.70 | 67.17 | 83.00 | 39.00 | 71.72 | 76.93 | 40.41 | 63.04 |
| | 8 | 3.58 | 65.75 | 85.00 | 41.00 | 71.34 | 76.39 | 40.48 | 63.33 |
| 20 | 4 | 2.85 | 67.88 | 84.00 | 40.20 | 72.35 | 77.69 | 41.25 | 63.90 |
| | 8 | 4.02 | 66.61 | 89.00 | 38.00 | 72.58 | 77.64 | 41.25 | 64.18 |
| 25 | 4 | 2.99 | 67.17 | 87.00 | 40.00 | 73.26 | 78.07 | 42.38 | 64.65 |
| | 8 | 4.46 | 68.67 | 86.00 | 41.00 | 73.00 | 78.67 | 41.79 | 64.86 |
| 30 | 4 | 3.14 | 69.69 | 89.00 | 40.60 | 73.92 | 77.53 | 42.82 | 65.59 |
| | 8 | 4.90 | 67.56 | 88.00 | 40.80 | 73.88 | 78.56 | 41.94 | 65.12 |

In Table 6, we compare the allocation of $\{4, 8\}$ bits of the selected top-$k$ experts, while all other experts are quantized to 2 bits. We quantize the shared experts and attention weights to 8 bits. Table 6 indicates that increasing the bit width of frequently activated experts improves performance. However, the gain from increasing the top-$k$ expert bits from 4 to 8 is minimal.

We summarize all experimental results and illustrate the relationship between bit width and average performance in Figure 6. Overall, we observe that as the bit width increases, the performance is improved. As highlighted by the red cross mark ✕ in the figure, achieving an average MoE bit width of 2.12 results in a performance score of 61.11, which marks a 5% improvement over the model quantized to 2 bits. This underscores the effectiveness of MoE blocks in settings with limited bit width.

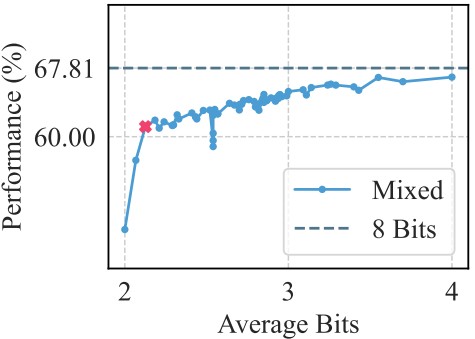

Figure 6: Performance of different quantization bits on `DeepSeek-MoE-16B-base` model.

**Combination of the weight outlier and expert usage frequency.** We conducted additional experiments on the `DeepSeek-MoE-16B-base` model by integrating bit-width allocation based on layers with significant weight outliers with allocation based on expert usage frequency to explore the trade-off between them. Specifically, we aimed for a total average bit budget of 2.97. We selected portions of the model to be quantized to 4 bits using a combination of the two heuristics, while quantizing all attention weights to 4 bits and all other weights to 2 bits. For selecting the 4-bit weights, we introduced a hyper-parameter, $\alpha$ ($0 < \alpha < 1$), representing the proportion of weights chosen based on expert usage frequency, with the remainder selected based on weight outliers. We varied $\alpha$ to illustrate the trade-off between these methods, as detailed above. As shown in Table 7, the optimal

combination of these two methods occurs when alpha is set to 0.1. This means that $20\%$ of the 4-bit MoE weights are selected based on expert usage frequency, while the remaining $80\%$ are chosen according to weight outliers.

Table 7: The combination of weight outlier and expert usage frequency, evaluated on the `DeepSeek-MoE-16B-base` model.

| Bits | $\alpha$ | WinoGrande (%) | COPA (%) | OBQA (%) | HellaSwag (%) | PIQA (%) | MMLU (%) | Average (%) |
|---|---|---|---|---|---|---|---|---|
| | 0.0 | 67.72 | 90.00 | 39.20 | 72.83 | 77.15 | 41.06 | 64.66 |
| | 0.1 | 68.11 | 89.00 | 41.60 | 72.88 | 77.80 | 41.84 | 65.21 |
| | 0.2 | 69.21 | 89.00 | 41.60 | 72.60 | 76.93 | 41.60 | 65.09 |
| | 0.3 | 68.92 | 88.00 | 42.00 | 72.06 | 76.65 | 41.21 | 64.81 |
| | 0.4 | 67.48 | 89.00 | 41.40 | 71.88 | 76.71 | 40.96 | 64.57 |
| 2.97 | 0.5 | 67.32 | 90.00 | 40.80 | 71.89 | 76.93 | 40.21 | 64.52 |
| | 0.6 | 65.90 | 87.00 | 39.40 | 71.86 | 76.76 | 38.67 | 63.27 |
| | 0.7 | 66.21 | 87.00 | 41.40 | 71.45 | 76.87 | 36.98 | 63.32 |
| | 0.8 | 66.45 | 89.00 | 41.00 | 70.89 | 76.60 | 37.67 | 63.60 |
| | 0.9 | 66.37 | 84.00 | 40.20 | 70.83 | 76.87 | 39.84 | 63.02 |
| | 1.0 | 68.19 | 87.00 | 41.60 | 71.01 | 76.11 | 40.81 | 64.12 |

**Baseline results of low-precision quantization.** We provide the 16-bit (FP16), 4-bit, and 2-bit baselines of both `Mixtral-8x7B` and `DeepSeek-MoE-16B-base` models in Table 8.

Table 8: Baseline results of the 16-bit (FP16), 4-bit, and 2-bit quantization.

| Bits | WinoGrande (%) | COPA (%) | OBQA (%) | HellaSwag (%) | PIQA (%) | MMLU (%) | Average (%) |
|---|---|---|---|---|---|---|---|
| | | | | `Mixtral-8x7B` | | | |
| 16 | 76.48 | 93.00 | 47.00 | 83.98 | 82.37 | 70.35 | 75.33 |
| 4 | 74.98 | 92.00 | 46.20 | 81.65 | 80.85 | 67.65 | 73.89 |
| 2 | 49.33 | 63.00 | 25.40 | 28.18 | 52.99 | 24.29 | 40.53 |
| | | | | `DeepSeek-MoE-16B-base` | | | |
| 16 | 70.40 | 91.00 | 44.20 | 77.35 | 78.72 | 44.77 | 67.74 |
| 4 | 71.35 | 87.00 | 43.20 | 76.39 | 78.51 | 44.22 | 66.78 |
| 2 | 53.28 | 76.00 | 30.20 | 45.33 | 66.54 | 25.28 | 49.44 |

