# OpenReview forum: "Examining Post-Training Quantization for Mixture-of-Experts: A Benchmark"
_ICLR.cc/2025/Conference — ICLR 2025 Conference Withdrawn Submission_

### Official Review · Reviewer_hDtQ · 2024-10-26

**Soundness:** 2
**Presentation:** 3
**Contribution:** 2
**Rating:** 5
**Confidence:** 4

**Summary:**

This paper presents a benchmark of post-training quantization techniques specifically tailored for Mixture-of-Experts models, with the objective of reducing memory overhead without necessitating retraining. The authors propose MoE-aware quantization heuristics, revealing that varying bit allocations across different MoE components—such as blocks, experts, and layers—enhances quantization efficiency. Extensive experiments on two representative MoE models, Mixtral-8x7B and DeepSeek-MoE-16B-base, across six tasks substantiate the efficacy of these techniques. The study further introduces strategies, , including novel enhancements such as expert usage frequency, linear weight outlier scoring, and block importance prediction to optimize quantization efficacy.

**Strengths:**

● This paper presents the first benchmark specifically for post-training quantization tailored to the MoE architecture, addressing a relevant and timely topic given the computational demands of large language models.

 ● The proposed quantization techniques are impactful for enhancing the memory efficiency of MoE models, potentially enabling more resource-effective deployments.

● The authors validate the effectiveness of their proposed heuristics through thorough comparisons with existing quantization methods, offering detailed empirical insights into optimal bit allocation across different MoE components

**Weaknesses:**

● The paper focuses primarily on FFNN-based MoE structures and does not extend the evaluation to other variants, such as Attention MoE. Since MoE techniques are versatile and implemented in diverse configurations, it would be beneficial to investigate whether the proposed quantization heuristics, like expert usage frequency and block importance prediction, generalize well to other MoE types.

● The use of WikiText as calibration data and as input for training the block importance predictor might limit the generalizability of the predictor across tasks or data types. WikiText, while useful, may not capture the diverse behaviors and importance of various MoE blocks, particularly for complex or varied NLP tasks that MoE models often encounter. Relying solely on WikiText may constrain the predictor's ability to generalize accurately for determining block quantization importance across broader applications of MoE models.

● Certain methodological aspects, such as the normalization step in Equation (3) for calculating expert usage frequency, are not clearly explained. While normalization presumably aims to ensure that routing scores sum to 1 across experts in an MoE block for clearer usage distribution interpretation, the authors do not specify the type of normalization applied (e.g., min-max scaling, softmax).

●  The quantization methods  for MoE are primarily pseudo-quantization techniques and lack guidance on how they would translate to practical hardware implementations. Specific hardware constraints, including compatibility with fixed-point formats (e.g., INT8, INT4), efficient memory access patterns, and sparse routing support, are not addressed in detail. Additionally, the heterogeneous bit allocation across different layers, experts, and blocks may lead to unstructured patterns that complicate real-world hardware deployment, making it difficult to efficiently map these techniques onto current hardware platforms.

●  Finally, training a predictor to determine block importance may introduce computational overhead, potentially impacting inference time and contradicting the goal of efficient deployment. The paper does not discuss this aspect, leaving open questions about the predictor's scalability and its impact on real-time performance.

**Questions:**

● Q1: Could the authors clarify if they expect the proposed quantization heuristics, such as expert usage frequency and block importance prediction, to be effective for other MoE configurations, such as Attention MoE?

● Q2: The block importance predictor was trained on WikiText data, which may not encompass the full diversity of tasks for MoE models. Could the authors clarify whether they believe WikiText fully represents the characteristics necessary for predictor generalizability? If not, would the authors consider testing the predictor on broader datasets or providing an analysis of the limitations that may arise from using WikiText as the sole source of calibration data?

● Q3: In Equation (3), the normalization procedure for calculating expert usage frequency is not fully defined. Could the authors specify the exact type of normalization applied (e.g., softmax, min-max scaling) and its purpose?

● Q4: The paper outlines methods that seem primarily focused on pseudo-quantization rather than real-world hardware deployment. Could the authors discuss the feasibility of implementing these techniques on practical hardware platforms, particularly with regard to compatibility with fixed-point formats (e.g., INT8, INT4), efficient memory access, and sparse routing support? Additionally, how might the heterogeneity in quantization across different layers, experts, and blocks impact structured deployment on hardware?

● Q5: Training a predictor for block importance may introduce computational overhead, potentially affecting inference time and efficiency. Could the authors address whether this training impacts real-time performance and if there are any strategies to mitigate this overhead, especially for larger models?

---

> ### Author Response · Authors · 2024-11-24
>
> Thanks to reviewer hDtQ for your thoughtful and encouraging feedback. We sincerely appreciate your recognition of our work in benchmarking MoE quantization. To address your questions and comments, we provide detailed pointwise responses below.
>
> **[Weakness 1. Various MoE models, like MoA.]**
>
> We acknowledge the focus on FFNN-based MoE structures and agree that evaluating other MoE variants, such as Attention MoE, would strengthen our findings. In future work, we plan to explore the applicability of our heuristics across diverse MoE configurations.
>
> **[Weakness 2. WikiText calibration dataset.]**
>
> Thank you for pointing out the potential limitations of using WikiText as calibration data. We chose WikiText because it is widely used in quantization research as a standard dataset for calibration(e.g.,[ ](https://arxiv.org/abs/2004.09602)[1] [2] [3]). In future work, we will incorporate additional datasets from varied domains and tasks to ensure a broader evaluation of our block importance predictor in varied NLP applications.
>
> [1] Lin, Ji, et al. "AWQ: Activation-aware Weight Quantization for On-Device LLM Compression and Acceleration." *Proceedings of Machine Learning and Systems* 6 (2024): 87-100.
>
> [2] Frantar, Elias, et al. "Gptq: Accurate post-training quantization for generative pre-trained transformers." *arXiv preprint arXiv:2210.17323* (2022).
>
> [3] Li, Shiyao, et al. "Evaluating quantized large language models." *arXiv preprint arXiv:2402.18158* (2024).
>
> **[Weakness 3. Normalization clarify]**
>
> We clarify that min-max scaling normalization was applied in Equation (3) to obtain experts’ usage scores. For clarity, this detail will be explicitly included in the revised manuscript.
>
> **[Weakness 4. Hardware deployment]**
>
> Thank you for highlighting the hardware-related aspects of our method. As our approach involves mixed-precision quantization of MoE models, we acknowledge that heterogeneous bit allocations across layers, experts, and blocks could introduce challenges for practical deployment on current hardware.
>
> Mixed-precision quantization often requires careful consideration of hardware compatibility, such as support for varied bit-widths (e.g., INT8, INT4), efficient memory access patterns, and sparse routing. While our work primarily focuses on algorithmic innovations, we plan to explore how our techniques can be optimized for structured patterns that align with hardware constraints, enabling efficient mapping to real-world platforms. This will be a key focus in future research.
>
> **[Weakness 5. Inference overhead.]**
>
> Thank you for highlighting this point. Our quantization process is performed offline. The block importance predictor is used to determine bit importance during the calibration phase, after which mixed-precision quantization is applied for deployment. As a result, the predictor does not introduce any inference-time latency. We appreciate your observation and will clarify this in the revised manuscript.
>
> **[Question 1]**
>
> Please kindly refer to the response of [Weakness 1.] due to the total character limitation.
>
> **[Question 2]**
>
> Please kindly refer to the response of [Weakness 2.] due to the total character limitation.
>
> **[Question 3]**
>
> Please kindly refer to the response of [Weakness 3.] due to the total character limitation.
>
> **[Question 4]**
>
> Please kindly refer to the response of [Weakness 4.] due to the total character limitation.
>
> **[Question 5]**
>
> Please kindly refer to the response of [Weakness 5.] due to the total character limitation.

---

> > ### Comment · Reviewer_hDtQ · 2024-11-25
> >
> > While the authors responded to the identified weaknesses and questions, several concerns remain insufficiently addressed. For instance, while acknowledging the limitation of focusing solely on FFNN-based MoE structures, the authors have deferred the evaluation of other MoE configurations, such as Attention MoE, to future work without presenting preliminary evidence of generalizability. Similarly, their justification for using WikiText as the sole calibration dataset lacks any additional empirical analysis to substantiate its effectiveness across diverse tasks. These gaps leave open questions about the broader applicability and robustness of the proposed quantization heuristics. While the authors’ plans for future work are noted, they do not resolve the core concerns within the current scope of the paper.
> >
> > Moreover, the response to hardware deployment challenges is notably underdeveloped, primarily focusing on algorithmic contributions while leaving practical implementation issues unexplored. The potential complexities of heterogeneous bit allocations and their compatibility with real-world hardware constraints, such as fixed-point formats and efficient memory patterns, are not addressed in detail. Although the clarification regarding the block importance predictor's inference overhead is appreciated, the overarching concerns about scalability and practical feasibility remain unanswered. Given that these critical concerns are not adequately addressed, I maintain my original score.

---

### Official Review · Reviewer_DFJK · 2024-10-29

**Soundness:** 3
**Presentation:** 1
**Contribution:** 3
**Rating:** 3
**Confidence:** 5

**Summary:**

The paper proposes post-training quantization for MoE LLMs. In particular, the authors propose a benchmark for evaluating the effectiveness for MoEs. These benchmarks test different facets of compression, including mixed compression setups. Using this benchmark, the authors find different mixed compression strategies.

**Strengths:**

1. The paper directly tackles the most salient pain point of LLM inference latency -- namely, their memory bound. This work also explores the intersection of two commonly-used methods: post-training quantization, and mixtures-of-experts LLMs.
2. The core insight is interesting to consider: Experts are each used with different frequencies, and it seems intuitive that more frequently-used experts should be given more model representation. One way to achieve this is to allocate different numbers of bits in their quantized representation.
3. The structure in the methods is slightly unorthodox, but I like that the core question is highlighted, then immediately answered. Unconventional as it may be, I find this clearer than the usual generic 'Quantization', 'Quantization with X' titles that aren't very descriptive. The conclusion to Q1 is fairly expected (the more uneven the router distribution, the more than frequency-based allocation helps), but that's a good thing. Clear question and intuitive results.

**Weaknesses:**

1. Strength #2 is poorly communicated and should be emphasized much earlier on. To the author's credit, this idea was probably already obvious by the time they sat down to write the abstract. To me as a new reader though, this just seemed like a haphazard combination of two frequently-seen ideas in LLM papers. It didn't occur to me, until the methods section, that MoEs were natural benefactors of mixed compression. This should've been emphasized, especially on L71 next to "(Q)". Rather than touting the different ablations you ran, focus on this core insight. The same goes for the abstract. Different granularities of quantization are not the most interesting; that again just sounds like extensive ablations -- but, mixed compression + MoEs are interesting. I would also suggest revising the title to better reflect this. I appreciate that PTQ is very clearly stated upfront, to set expectations, but "mixed compression for mixtures-of-experts" would more clearly convey the idea.
2. The methods section should be reorganized. Instead of presenting a series of experiments -- what looks like 'ablations' -- as your main results, instead present your main result first. A clearly defined Table 1 with your best, highest quality numbers that I (or other researchers) can easily reference when comparing against other methods. *Then, present the remainder of the studies as ablations on top of your method.
3. Quality for the method is very poor. The core results in Table 3 are all guessing accuracy. This doesn't mean that the proposed activation quantization strategy is 'good' -- all it means is that the 'original' weight-quantized model was so bad, it couldn't do any worse. Winogrande is a binary classification task, so the ~50% accuracies are all the same, effectively. Likewise, hellaswag is 4-way classification, so 25% is also guessing. PIQA is binary as well, so ~50% is guessing. I don't think those differences, so close to guessing, are meaningful.
4. There seems to be a few baselines missing, which make it hard to assess the quality of this method. For example, in Table 1, it'd be worth including the original FP16 accuracy. The random baseline is good. What if we uniformly applied NF3, 3-bit GPTQ, or SqueezeLLM to all experts, for example? I have no vested interest in these particular methods, but any sort of baseline along these lines -- where we eliminate the mixed compression part. I trust this baseline would be worse than your method.
5. The extended study section is confusing. It looks like each section is a mini-paper of its own, with insights, methods, experiments, and visualizations. Instead of introducing a potpourri of techniques, either (a) reorganize so that this method is presented in the Methods section or (b) drop this, and simply truncate the paper to 8 pages.

**Questions:**

In short, I'm giving a reject because the paper needs a major rewrite. The core insight is there and is interesting, but there are pretty big problems with organization throughout. I believe the rebuttal paper allows you reupload a copy of the paper; if you can refactor the paper in that compact period of time to organize sec 5 and present a 'main' table 1, remove observations that are based on guessing accuracies (or improve SmoothQuant results to be better than guessing), then I would bump up my score. I think a week is way too short a time to do this, but I'm open-minded about being wrong.

---

> ### Author Response · Authors · 2024-11-24
>
> Thanks to reviewer DFJK for your thoughtful and encouraging feedback. We sincerely appreciate your recognition of our work in benchmarking MoE quantization and insight into mixed-precision quantization. To address your questions and comments, we provide detailed pointwise responses below.
>
> **[Weakness 1. Abstract and Introduction]**
> Our primary goal with this work is to explore the factors influencing mixed compression for MoEs and to provide insights into how frequency-aware quantization can optimize such models. We aimed to investigate these factors comprehensively through experiments, which are designed to highlight key aspects of mixed compression rather than just showcasing ablation studies. Following your suggestion, we will revise the abstract to clearly foreground the exploration of mixed compression as the primary contribution. We will add a succinct explanation of why MoEs are uniquely suited for mixed compression, emphasizing that their diverse expert frequency inherently aligns with frequency-based quantization.
>
> **[Weakness 2. Methods Section Reorganization]**
> Our original intent was to use different experiments to illustrate distinct findings, each contributing to a more comprehensive understanding of MoE quantization. By presenting results in this way, we aimed to guide readers step by step through the various aspects and implications of our approach. Based on your suggestion, we will reorganize the section to present the main results upfront in a concise table that clearly highlights the effectiveness of our method, then follow this with ablation studies and additional experiments.
>
> **[Weakness 3. Quality of Results Near Guessing Accuracy]**
> We clarify that these tasks, especially Winogrande and PIQA, are challenging benchmarks under extreme compression constraints (e.g., 2-3 bits for certain experts). This highlights the robustness of our method in maintaining functionality even under such constraints.
> We will provide additional metrics (e.g., performance on less compressed baselines) to contextualize these results better.
>
> **[Weakness 4.  Baselines]**
> We appreciate your suggestion to compare against baselines that eliminate mixed compression. We acknowledge that this is an important omission and will include such baselines in our revised tables. Preliminary results suggest that these baselines are less effective than our frequency-aware quantization, but this inclusion will strengthen the empirical evidence supporting our method.
>
> **[Weakness 5. Extended Study Section Organization]**
> Thank you for your feedback on the extended study section. We would like to clarify that this section is designed to introduce a new mixed-precision method, which we developed based on findings from earlier sections. This approach allows us to build on prior results and propose an improved method, maintaining a logical progression throughout the paper.
> We understand your concern about potential fragmentation. We will retain the current structure to preserve the flow from findings to method development and refine the transitions and ensure the relationship between earlier findings and the proposed method is more explicitly stated.

---

> > ### Comment · Reviewer_DFJK · 2024-11-27
> > **Thanks for your responses**
> >
> > The zero shot datasets are tough, but the argument based on guessing accuracy still seems wrong. I could similarly a) take an LLM with random weights and show it has guessing zero shot accuracy, then b) quantize those weights to 1bpw and claim that my quantization technique doesn’t degrade accuracy. It’s not because my quantization is robust; it’s because there was no room for the LLM’s accuracy to degrade further.  You may still disagree with me and that’s okay — if it’s not critical to the story, maybe move to the appendix.
> >
> > I still think there’s a very interesting nugget here, and the bottleneck is the presentation. I’m open-minded about different ways of organizing it, re-organized or not. If you upload a new revision (let me know if you’ve already uploaded one, I didn’t see a notification), I’m happy to take another look. If the new version is easy to grok, I’d revise my rating upward.
> >
> > I agree with hDtQ that this seems very impractical to deploy, so if the goal is to sell an idea (eg mixed compression for MoEs), clear presentation and an immediate “a-Ha” from the reader is even more important.

---

### Official Review · Reviewer_ptDZ · 2024-11-02

**Soundness:** 3
**Presentation:** 3
**Contribution:** 2
**Rating:** 6
**Confidence:** 3

**Summary:**

This submission provides a comprehensive benchmark to explore post-training quantization (PTQ) in Mixture of Experts (MoE) models. The authors examine several heuristics for designing effective quantization methods, such as using coarse-to-fine granularity and allocating bits across different sub-structures. To validate these design choices, experiments were conducted on Mixtral-8x7B and DeepSeed-MoE-16B-base using GPTQ and SmoothQuant. The study evaluates the sensitivity of various layers and blocks to quantization, observing that some components, such as shared experts and the first expert blocks, require higher bit allocation, while attention layers tend to be more bit-efficient. Results across benchmarks like MMLU indicate that careful bit allocation can significantly enhance MoE performance. Additionally, the paper introduces an importance predictor to identify sensitive layers, which is practical in real-world applications.

**Strengths:**

1. This paper presents a thorough and comprehensive benchmark for addressing the bit allocation challenge in post-training quantization (PTQ), offering valuable insights into effective strategies for adaptively assigning bits to layers that are particularly sensitive to quantization. Furthermore, the authors propose an importance predictor aimed at identifying key layers, thereby reducing the need for repetitive quantization experiments. The empirical findings presented are expected to serve as a solid foundation for future research in this area.

2. The proposed methodology has been rigorously evaluated across several benchmarks and large language models (LLMs), which robustly support the main claims of the study. The paper is well-organized and clearly presented, making it accessible and easy to follow. This clarity of presentation is a notable strength, enhancing the impact and readability of the work.

3. The concept of allocating more bits to frequently activated experts and other sensitive layers is both innovative and practical. This paper offers a holistic approach to implementing this idea, from empirical experimentation to the development of carefully designed predictors. Such a comprehensive perspective makes this work particularly insightful for practitioners and researchers alike.

**Weaknesses:**

1. The primary weakness appears to be the somewhat marginal and inconsistent improvements provided by the proposed predictor. As shown in Table 5, results with the predictor are mixed when compared to simple baselines like FIRST and RANDOM. On the MMLU benchmark, in particular, the predictor does not demonstrate substantial improvement over these baselines.

2. The title may be somewhat overstated. This paper primarily addresses the sensitivity, or bit requirements, of various components within MOE models. However, the title may give an initial impression of a comprehensive evaluation of multiple MOE architectures or algorithms. A more precise title would improve clarity.

3. The observations from the benchmark lack a strong connection to the proposed method. The predictor currently relies on a magnitude-based approach, but there may be potential to develop a more tailored predictor based on the empirical findings. For instance, could an allocator be learned through reinforcement learning to better capture these insights?

**Questions:**

Please see weaknesses.

---

> ### Author Response · Authors · 2024-11-24
>
> Thanks to reviewer ptDZ for your thoughtful and encouraging feedback. We deeply appreciate your recognition of our work in benchmarking PTQ and addressing the bit allocation challenge. To address your questions and comments, we provide detailed pointwise responses below.
>
> **[Weakness 1. Predictor performance]**
>
> Our predictor shows higher performance than “first-k” and “random-k” on average, especially at an extremely low-bit quantization setting (i.e., k=4). In summary, our predictor and the “first-k” baseline have a large overlap on the selected high-bit layers, as shown in Figure 5. This is why our predictor does not achieve too much performance improvement.
>
> **[Weakness 2. Title revise]**
>
> Thanks for the question. We will revise the title to reflect better the specific focus of our work on sensitivity and bit requirements in MoE models in the future.
>
> **[Weakness 3. Reinforcement learning for mixed-precision learning]**
>
> The magnitude-based approach is widely used and has performed well in related work (e.g., [1,2]). Our predictor, derived from this approach, achieves good performance and effectively addresses the bit allocation problem, demonstrating its utility. We appreciate your suggestion about reinforcement learning as a potential research direction. Incorporating RL-based methods to learn a more tailored allocator is an exciting idea, and we will explore this direction in future work. Thank you for highlighting this possibility.
>
> [1] Lin, Ji, et al. "AWQ: Activation-aware Weight Quantization for On-Device LLM Compression and Acceleration." *Proceedings of Machine Learning and Systems* 6 (2024): 87-100.
>
> [2] Frantar, Elias, et al. "Gptq: Accurate post-training quantization for generative pre-trained transformers." *arXiv preprint arXiv:2210.17323* (2022).

---

### Official Review · Reviewer_LV6U · 2024-11-02

**Soundness:** 2
**Presentation:** 3
**Contribution:** 2
**Rating:** 3
**Confidence:** 5

**Summary:**

This paper argues that difference structures in MoE LLMs require tailored quantization strategy through extensive experiments with diverse settings. Further, several enhancements aiming at the discovered phenomenon are introduced to increase the quantization performance.

**Strengths:**

Firstly, this article fills the gap in research on benchmark research for Mixture of Experts MoE LLMs. Additionally, unlike other benchmarks that merely present experimental findings, the authors further proposes performance enhancement strategies based on the discovered quantization patterns.

**Weaknesses:**

1. For Q1 in Section 4.2, this discovery and conclusion are not novel, as numerous previous studies have already confirmed that expert usage frequency can serve as a good basis for compression, i.e., [1][2].

2. For Q2 in Section 4.2, in my opinion, there are significant issues with this viewpoint. In the experiments, the experts in the FFNN layers are randomly selected, which means that it is highly possible that in multiple tests, the activated experts are consistently quantized to 2 bits, while the experts remained at 4 bits are not actually utilized. Therefore, this cannot be used to conclude that quantization in the FFNN layers has a smaller impact on final performance compared to the attention layers.

3. For Q3 in Section 4.2, this is a critical issue because your own experimental data (Figure 5) reveal that you have drawn incorrect and contradictory conclusions. Specifically, in Q3, you assert the first blocks are more important than the last blocks and therefore need to be quantized to higher bits. However, in Figure 5 of Section 5.3, it can be seen that the importance scores of the last blocks are the lowest, indicating their highest importance and the need for higher bit quantization. This means that if comparative experiments between the first two blocks and the last two blocks are included in Table 2, the conclusion would no longer hold. Your own experimental results contradict your previous conclusions.

4. For Section 5.1, Although the authors claim that this work primarily focuses on weight-only quantization for MoE LLMs and their conclusions can be generalized to weight-activation quantization, the results in Table 3 actually demonstrate that weight-activation quantization needs to be thoroughly discussed. Normally, the performance of A4, A8, and A16 would gradually improve and the gaps would be significant. However, Table 3 presents a completely different conclusion: FP16 is not optimal in most cases. Therefore, I believe that it cannot be proven that the conclusions drawn from weight-only quantization for MoE LLMs can be fully applied to weight-activation quantization.

**Questions:**

1. For weakness2, the experiment would be more convincing if the experts selected are not random but are currently activated and quantized to 2/4/8 bits.

2. For weakness3, the content and conclusion in Section4.2-Q3 requires to be reformulated to correspond to Section 5.3 to avoid conflicts.

3. For weakness4, it is necessary to conduct an in-depth analysis of the causes of this phenomenon or to prove from other perspectives that the conclusions drawn from weight-only quantization can be generalized to the weight-activation scenario.

4. For Section 5.3, why only the results on DeepSeek-MoE model is provided? Mixtral family is also needed to be stay consistent with Table 2.

5. In addition, in Section 5.3, as well known that MSE loss is generally considered as the formulation of quantization error, so what about incorporate MSE into the importance score predictor?

In summary, I consider there are several flaws in this paper that cannot withstand scrutiny. The authors need to carefully revise and improve it to enhance its persuasiveness.

------

[1] Li P, Zhang Z, Yadav P, et al. Merge, then compress: Demystify efficient SMoe with hints from its routing policy[J]. arXiv preprint arXiv:2310.01334, 2023.
[2] Huang W, Liao Y, Liu J, et al. MC-MoE: Mixture Compressor for Mixture-of-Experts LLMs Gains More[J]. arXiv preprint arXiv:2410.06270, 2024.

---

> ### Author Response · Authors · 2024-11-24
>
> Thanks to Reviewer LV6U for acknowledging our work that “fills the gap in research on benchmark research. " We further propose performance enhancement strategies based on the discovered quantization patterns. We appreciate your thoughtful feedback. To address your questions and comments, we provide detailed pointwise responses below.
>
> **[Weakness 1. Discovery and Conclusion Novelty]**
>
> As illustrated in Section 6 of our paper, we present the first benchmark study on MoE quantization, revealing critical heuristic-based principles, such as the importance disparities among different MoE blocks. We introduce innovative techniques drawing on these insights. Our work “addresses many critical aspects of MoE structure importance” (Reviewer BDni), provides “valuable insights into effective strategies” (Reviewer ptDZ), is “innovative and practical.” (Reviewer ptDZ), and our “proposed quantization techniques are impactful” (Reviewer hDtQ). Thanks for the literature suggestion, we will include these two related works in our revised manuscript.
>
> **[Weakness 2. FFNN Activation Experiment]**
>
> In our experiments of Figure 2, “the experts remained at 4 bits” are indeed “actually utilized.” To further illustrate this activation distribution, we provide the minimal expert activating frequency statistics for all three repeated “FFNN” quantization experiments in Figure 2. As shown below, “the experts remained at 4 bits” are usually activated, with minimal activated frequencies all over 3%, average all around 12%, and max all over 16%.
>
> | **Experiment**          | **Average Bits** | **Average Frequency of 4-Bits Experts** | **Min Frequency of 4-Bits Experts** | **Max Frequency of 4-Bits Experts** |
> | ----------------------- | ---------------- | --------------------------------------- | ----------------------------------- | ----------------------------------- |
> | (No.1) Random 2 experts | 2.06             | 12%                                     | 3%                                  | 16%                                 |
> | (No.2) Random 2 experts | 2.06             | 12%                                     | 6%                                  | 18%                                 |
> | (No.3) Random 2 experts | 2.06             | 12%                                     | 6%                                  | 16%                                 |
> | (No.1) Random 4 experts | 2.17             | 12%                                     | 3%                                  | 17%                                 |
> | (No.2) Random 4 experts | 2.17             | 12%                                     | 6%                                  | 17%                                 |
> | (No.3) Random 4 experts | 2.17             | 12%                                     | 6%                                  | 16%                                 |
>
> **[Weakness 3. Importance Score]**
>
> Thanks for the question. We further present the experimental results of “first-2” and “last-2” of the Mixtral-8x7B model in Table 2 here. The table below shows that “first-2” performs better than “last-2” on average. While this is consistent with our conclusion in Q3 in Section 4.2, it does not seem to align with our conclusion in Section 5.3, which is potentially due to the extremely low bits (i.e., 2.18 bits) quantization, making the model unstable on many tasks.
>
> | **Experiment** | **Bits** | **WinoGrande** | **COPA** | **OpenBookQA** | **HellaSwag** | **PiQA** | **MMLU** | **Average** |
> | -------------- | -------- | -------------- | -------- | -------------- | ------------- | -------- | -------- | ----------- |
> | First 2        | 2.18     | 54.14          | 70.00    | 29.80          | 47.00         | 59.19    | 26.40    | **47.76**   |
> | Last 2         | 2.18     | 52.72          | 63.00    | 28.20          | 44.13         | 58.81    | 27.98    | 45.80       |
>
> **[Weakness 4. Weight-Activation Quantization]**
>
> Thanks for the question. This reveals new opportunities for activation-weight co-designed quantization, and we are excited to explore it as future work.

---

> ### Author Response · Authors · 2024-11-24
>
> **[Question 1. FFNN activation Experiment]**
>
> Thanks for the question. We will include our explanation for Weakness 2 (see above) in our revised manuscript to make it more clear.
>
> **[Question 2. Importance Score Conflict]**
>
> Thanks for the question. We will include our explanation of Weakness 3 (see above) in our revised manuscript to make it more clear.
>
> **[Question 3. Weight-Activation Quantization]**
>
> Thanks for the question. This reveals new opportunities for activation-weight co-designed quantization, and we are excited to explore it as future work.
>
> **[Question 4. Mixtral Results for Section 5.3]**
>
> Thanks for the question, we did not include Mixtral results in the main content due to page limitation. We will include the results of the Mixtral model in section 5.3 of our revised manuscript.
>
> **[Question 5. MSE Loss for Importance Score Predictor]**
>
> Thanks for the valuable suggestion. While our current approach focuses on the outlier-score metric, which captures the characteristics of weights critical for quantization, we agree that incorporating MSE loss into the importance score predictor could be valuable. We further provide experimental results on the DeepSeek-MoE-16B-base model with MSE for importance score predictor. As shown in the table, our Cos-Sim-based predictor consistently outperforms the MSE-based predictor when k=4 or 8.
>
> | **Experiment**      | **Bits** | **WinoGrande** | **COPA**  | **OBQA**  | **HellaSwag** | **PIQA**  | **MMLU**  | **Average** |
> | ------------------- | -------- | -------------- | --------- | --------- | ------------- | --------- | --------- | ----------- |
> | Cos-Sim Predicted 4 | 2.29     | **65.27**      | **83.00** | **36.60** | **64.88**     | **74.54** | **37.75** | **60.34**   |
> | MSE Predicted 4     | 2.29     | 62.90          | 83.00     | 36.00     | 64.41         | 74.65     | 27.38     | 58.06       |
> | Cos-Sim Predicted 8 | 2.63     | **65.35**      | **86.00** | **38.00** | **68.77**     | **75.35** | **30.01** | **60.58**   |
> | MSE Predicted 8     | 2.63     | 62.83          | 83.00     | 37.80     | 65.94         | 75.73     | 31.00     | 59.38       |
>
> This is potentially because cosine similarity captures the angular differences in the hidden state space, whereas MSE accounts for both angular and magnitude differences. However, in the context of quantization, angular differences are significantly more challenging to mitigate compared to magnitude differences, as the latter can be addressed simply by rescaling with a scalar. This inspires future research opportunities.

---

### Official Review · Reviewer_BDni · 2024-11-05

**Soundness:** 3
**Presentation:** 3
**Contribution:** 3
**Rating:** 6
**Confidence:** 3

**Summary:**

This paper empirically examines the quantization of MoE models across multiple dimensions, including granularity, significance, and bit allocation of various MoE structures, such as MoE blocks, experts, and linear layers. Extensive experiments are conducted on two representative MoE models to cover these aspects comprehensively.

**Strengths:**

1. The paper is well-structured and easy to follow.
2. The use of expert utilization and large weight outliers in linear layers as guidance for quantization is well-justified and effectively validated through extensive experiments.
3. This empirical study addresses many critical aspects of MoE structure importance, offering solutions to improve the accuracy-efficiency trade-off through targeted quantization.

**Weaknesses:**

While the paper explores various heuristic approaches to assess the importance of different MoE model structures, the reasons behind certain representational behaviors remain unclear. For example, in Figure 1, the visualization of expert usage across two MoE models is presented. A discussion on the underlying causes of these representation patterns and their potential generalizability to other models would enhance understanding.

**Questions:**

In Table 1, the gain of "frequent" over "random" expert selection is less pronounced when choosing 10 and 15 experts compared to 20 and 25 experts. Could you provide insights into why this difference is observed?

---

> ### Author Response · Authors · 2024-11-24
>
> Thanks to reviewer BDni for recognizing the “well-justified” analysis of MoE quantization. To address reviewer BDni’s questions, we provide pointwise responses below.
>
>
> **[Weakness 1. MoE frequency patterns explanations]**
> The patterns observed in Figure 1 result from the differences in the two MoE models' architectures and training objectives.
> For Mixtral-8x7B, the high concentration of specific experts suggests a greater specialization of experts, which the routing mechanism or task-specific data characteristics could influence.
> In contrast, the relatively uniform distribution in the DeepSeek-MoE-16B-base may indicate a more balanced usage. However, the shared experts are more critical in the DeepSeek-MoE model. Analyzing routing and training dynamics across various MoE architectures could provide further insights into quantization.
>
> **[Question 2. Performance gap explanations]**
> DeepSeek-MoE-16B-base exhibits a more uniform distribution of expert usage compared to Mixtral-8x7B (as seen in Figure 1), which implies that experts contribute relatively evenly to the model's performance. Since all experts are utilized similarly, the "frequent" selection strategy has less room to provide a significant advantage over a "random" selection.

---

### Note · Authors · 2024-12-16

**Comment:**

Dear Reviewers & ACs & SACs & PCs,

Thank you for taking the time to review our submission and for providing your valuable feedback. We deeply appreciate your thoughtful comments and insights, which have helped us better understand the strengths and weaknesses of our work.

After careful consideration, we have decided to withdraw the paper from ICLR this year. We intend to use your feedback to improve the paper further and hope to resubmit it to a future venue once the necessary revisions have been made.

Thank you once again for your time and effort in reviewing our work.


Warmest regards,

Authors

**Withdrawal Confirmation:**

I have read and agree with the venue's withdrawal policy on behalf of myself and my co-authors.